# Total small vessel disease burden and functional outcome in patients with ischemic stroke

Wi-Sun Ryu[ID][1,2]*, Sang-Wuk Jeong[1,2], Dong-Eog Kim[1,2]

**1** Department of Neurology, Dongguk University Ilsan Hospital, Goyang, Republic of Korea, **2** Korean Brain MRI Data Center, Goyang, Republic of Korea

* wisunryu@gmail.com

## Abstract

### Background

Cerebral small vessel disease (SVD) is comprised of lacunes, cerebral microbleeds (CMBs), white matter hyperintensities (WMHs), and enlarged perivascular space (EPVS). We investigated the cumulative effect of SVD on 3-month functional outcome following ischemic stroke using the total SVD score.

### Methods

The total SVD score of 477 acute ischemic stroke patients with adequate brain MRI was analyzed. We used multivariable ordinal logistic regression analysis to investigate the independent impact of total SVD score on ordinal modified Rankin Scale (mRS) score at 3-month after ischemic stroke.

### Results

Mean age was 66±14 years, and 61% were men. The distribution of the total SVD score from 0 to 4 was 27%, 24%, 26%, 16%, and 7%, respectively. The proportion of mRS scores 2 or greater was 16% and 47% in total SVD score 0 and 4, respectively. Multivariable ordinal logistic regression analysis results showed that compared with the total SVD score of 0, total SVD scores of 2, 3, and 4 were independently associated with higher mRS scores with adjusted odds ratios (95% confidence intervals) of 1.68 (1.02–2.76), 2.24 (1.25–4.00), and 2.00 (1.02–4.29). Lacunes, CMBs, WMHs but not EPVS were associated with mRS scores at 3 months. However, the impact of each SVD marker on stroke outcome was smaller than that of the total SVD score.

### Conclusion

We found an independent association between total SVD scores and functional outcome at 3 months following ischemic stroke. The total SVD score may be useful for stratification of patients who are at a high-risk of unfavorable outcomes.

**Data Availability Statement:** Data cannot be shared publicly due to ethical restrictions. Data are available from the Institutional Review Board of Dongguk University Ilsan Hospital for researchers who meet the criteria for access by review of our

Institutional Review Board. For more information about data (DUIH-Korea) access, please contact the Korean Brain MRI Data Center at Dongguk University Ilsan Hospital (lcy815@hanmail.net).

**Funding:** WSR: This work was supported by the Dongguk University Research Program. The funders had no role in study design, data collection and analysis, decision to publish, or preparation of the manuscript.

**Competing interests:** The authors have declared that no competing interests exist.

## Introduction

Cerebral small vessel disease (SVD) is frequently observed in MRIs of ischemic stroke patients [1]. White matter hyperintensities (WMH), lacunes, cerebral microbleeds (CBMs), and enlarged perivascular space (EPVS) are neuroimaging markers of SVD [2]. Accumulating evidence has suggested that SVD is associated with incident stroke, recurrent stroke, and vascular cognitive impairment [1, 3]. In addition, SVD portends unfavorable functional outcome after ischemic stroke [4–6]. This has been shown in a recent large-sample sized study, patients with the highest quintile of WMHs had a nearly two-fold increased risk of higher modified Rankin Scale (mRS) score at 3-months than those in the lowest quintile [6].

In general, prior studies have reported on the clinical implication of individual SVD markers after ischemic stroke [6–9]. However, in clinical practice, SVD markers are often simultaneously present in a single patient. Hence, the cumulative effect of SVD markers has been recently highlighted, and a simple SVD score has been proposed and validated [10, 11]. A few studies have made observations on the impact of SVD score on functional outcome after ischemic stroke [12–14]. However, these studies are limited because they used incomplete SVD score (a combination of WMH and lacunes, or exclusion of CBMs). In the present study, we investigated the effect of the cumulative SVD burden on the 3-month functional outcome after ischemic stroke using the total SVD score.

## Methods

### Study population

We performed post hoc analysis on prospectively-collected data from a single-center stroke registry. From February 2012 to September 2013, we included a total of 687 patients who visited our center within 7 days of stroke onset and were confirmed to have had ischemic stroke based on brain MRI. The institutional review board of Dongguk University Ilsan Hospital approved the study protocol, and all participants gave written informed consent. The study protocol was conducted in accordance with the Declaration of Helsinki.

### Clinical data and outcome measurement

National Institutes of Health Stroke Scale (NIHSS) score at admission, pre-stroke mRS score, and mRS score at three months after stroke were collected prospectively. Under a standardized protocol [6, 15], we collected demographic data, prior medication history, laboratory data, past medical history including risk factors for stroke such as hypertension, diabetes mellitus, hyperlipidemia, coronary artery disease, atrial fibrillation, and smoking history. Stroke subtypes were determined by the consensus of experienced neurologists, using a validated MRI-based algorithm [16].

### MRI scoring

We obtained MRI with a 1.5 T or 3.0 T MR scanner, and the images included standard axial T2–weighted fast spin echo-images, axial fluid-attenuated inversion recovery (FLAIR) images, and T2-weighted gradient echo-images. A stroke neurologist (W. R.) blinded to the clinical data provided ratings for all of the obtained imaging. In accordance with the STRIVE guidelines [2], we defined lacunes as rounded or ovoid lesions, 3–15 mm in diameter, located in the basal ganglia, internal capsule, centrum semiovale, or brainstem, of CSF signal intensity on T2 and FLAIR, generally with a hyperintense rim on FLAIR and no increased signal on diffusion-weighted image (DWI). We quantified WMH volume, as previously described [17, 18]. In brief, brain template images ($1 \times 1 \times 1$ mm$^3$ voxel) were chosen from the Montreal

Neurological Institute(MNI) template within the range of -63.5 to 74.5 mm in the Z-axis of the MNI coordinate space. After normalization of images, each patient's high signal intensity lesions on FLAIR were semi-automatically segmented and registered onto the brain templates under careful supervision by a vascular neurologist (W. R.). When chronic lesions on FLAIR and acute lesions on DWI overlapped, the extent and distribution of FLAIR WMH contralateral to the location of acute infarct served as a reference to determine what volumes to include and exclude, by assuming the symmetric distribution of WMHs across the midline. We calculated the WMH volume on FLAIR as a percentage of total brain volume by dividing the number of voxels in the lesions over the total number of brain voxels, with corrections applied to account for the differences in scan slice thicknesses by adjusting the denominators. We defined EPVS as round, oval, or linear-shaped lesions with a smooth margin, absence of mass effect, and with a signal intensity equal to CSF on T2-weighted images and hypointense on FLAIR images without a hyperintense rim to distinguish them from old lacune. We counted EPVS at the level of the centrum semiovale and basal ganglia with a validated four-point visual rating scale (0 = none; 1 = 1–10; 2 = 11–20; 3 = 21–40; and 4 = >40) [19]. We counted enlarged perivascular spaces on the slide with the highest number in one hemisphere. We defined CMBs as rounded hypointense lesions on T2-weighted gradient echo-images with a diameter ≤ 10 mm. To calculate the total SVD score, we used a 5-point ordinal scare developed by Klarenbeek et al. [11]. A point was awarded if one or more lacunes were present, or WMH was found to be above the median of WMH volume, or one or more CMBs were present, or EPVS was found to be of moderate to severe degree (scored 2–4). Hence, the total SVD score ranged from a minimum score of 0 to a maximum of 4.

## Statistical analysis

Data are presented as mean [standard deviation (SD)], median [interquartile range (IQR)], and number (percentage), as appropriate. We compared baseline characteristics between total SVD scores using the chi-square test and ANOVA. For the skewed continuous variable, we used the Kruskal-Wallis test. To examine the distribution of the mRS score by total SVD score, we used the chi-square test. To investigate whether there is an independent association between total SVD score and functional outcome after ischemic stroke, we used multivariable ordinal logistic regression analysis with mRS scores as an ordinal outcome variable. Predefined covariates potentially associated with functional outcome after ischemic stroke, including age, sex, pre-stroke mRS score, initial NIHSS scores, previous history of stroke, hypertension, diabetes, smoking, coronary artery disease, stroke subtypes, and revascularization, were entered into the multivariable model. Data were analyzed using STATA 16.0 software (StataCorp. 2019. Stata Statistical Software: Release 16. College Station, TX: StataCorp LLC), and $p < 0.05$ were considered statistically significant.

## Results

Among 687 patients who were screened, we excluded those with the following to calculate the total SVD score; patients with incomplete MRI sequences(n = 135), poor MRI quality (n = 32), and lost to follow-up (n = 43), leaving 477 patients for the analyses. The mean age was 66.3 (standard deviation 14.1), 38% were women (n = 183), the median NIHSS 3 (interquartile range 1–5), and the median WMH volume was 0.85% of total brain volume (interquartile range 0.39–1.80). Prevalences of lacunes, CMBs, and EPVS were 53%, 26%, and 56%, respectively. The distribution of the total SVD score from 0 to 4 was 27%, 24%, 26%, 16%, and 7%, respectively. The patients with higher total SVD scores were older and had a history of hypertension compared to those with lower total SVD scores (Table 1). Previous stroke history and hyperlipidemia were more prevalent in patients with higher total SVD scores than those with

**Table 1. Baseline characteristics and total small vessel disease score.**

| | All patients (N = 477) | Small vessel disease score (N = 477) | | | | | p value |
|---|---|---|---|---|---|---|---|
| | | 0 (n = 130) | 1 (n = 114) | 2 (n = 124) | 3 (n = 77) | 4 (n = 32) | |
| Age, years | 66±14 | 58±15 | 65±13 | 70±13 | 72±10 | 76±10 | 0.01 |
| Sex, men | 294 (62%) | 81 (62%) | 75 (66%) | 81 (65%) | 42 (55%) | 15 (47%) | 0.19 |
| Pre-stroke mRS 2 or more | 60 (13%) | 4 (5%) | 11 (10%) | 19 (15%) | 12 (16%) | 12 (38%) | < 0.001 |
| Previous stroke | 99 (21%) | 9 (10%) | 19 (17%) | 32 (26%) | 26 (34%) | 13 (41%) | < 0.001 |
| Hypertension | 359 (75%) | 77 (59%) | 84 (74%) | 100 (81%) | 68 (88%) | 30 (94%) | < 0.001 |
| Diabetes | 182 (38%) | 41 (32%) | 43 (38%) | 54 (44%) | 30 (39%) | 14 (44%) | 0.36 |
| Hyperlipidemia | 143 (30%) | 27 (21%) | 36 (32%) | 36 (29%) | 32 (41%) | 12 (38%) | 0.024 |
| Smoking, current or quit < 5years | 221 (46%) | 61 (47%) | 58 (51%) | 60 (48%) | 31 (40%) | 11 (34%) | 0.39 |
| Coronary artery disease | 75 (16%) | 16 (12%) | 14 (12%) | 23 (19%) | 16 (21%) | 6 (19%) | 0.33 |
| Stroke subtype* | | | | | | | 0.12 |
| Large artery disease | 214 (45%) | 61 (47%) | 47 (41%) | 55 (44%) | 36 (47%) | 15 (47%) | |
| Small vessel occlusion | 103 (22%) | 19 (15%) | 26 (23%) | 28 (23%) | 23 (30%) | 7 (22%) | |
| Cardioembolism | 73 (15%) | 17 (13%) | 20 (18%) | 23 (19%) | 11 (14%) | 2 (7%) | |
| Undetermined | 65 (14%) | 22 (17%) | 16 (14%) | 15 (12%) | 6 (8%) | 6 (19%) | |
| Other-determined | 21 (4%) | 11 (9%) | 5 (4%) | 3 (2%) | 0 | 2 (6%) | |
| NIHSS, median (IQR) | 3 (1–5) | 2 (1–4) | 2 (1–4) | 3 (1–5) | 3 (1–5) | 4 (2–10) | 0.009 |
| WMH volume†, median (IQR) | 0.85 (0.39–1.80) | 0.32 (0.21–0.52) | 0.61 (0.35–1.22) | 1.11 (0.68–1.84) | 1.84 (1.11–2.75) | 3.47 (2.70–4.17) | <0.001 |
| Presence of lacunes | 251 (53%) | 0 | 38 (33%) | 102 (83%) | 76 (99%) | 32 (100%) | <0.001 |
| Presence of CMBs | 125 (26%) | 0 | 7 (6%) | 30 (25%) | 56 (74%) | 32 (100%) | <0.001 |
| Presence of EPVS | 268 (56%) | 0 | 66 (58%) | 98 (79%) | 72 (94%) | 32 (100%) | <0.001 |

The following values of the presented data are shown: mean±standard deviation, number (percentage), or median (inter-quartile range).

*Data are missing for one patient.

†Percentage of total brain volume.

mRS indicates modified Rankin Scale score; IQR, interquartile range; WMH, white matter hyperintensity; CMB, cerebral microbleeds; EPVS, enlarged perivascular space.

lower total SVD scores. In addition, NIHSS scores increased along with total SVD scores. Each SVD marker was associated with other markers of SVD (Fig 1). Compared with patients without lacunes, those with lacunes had more prevalent CBMs, above median of WMH, and EPVS (all P < 0.001). This association was observed in CMBs, WMH, and EPVS.

Fig 2 demonstrated the distribution of mRS score at 3-month after ischemic stroke stratified by total SVD scores. As the total SVD score increased, more patients had high mRS scores (P < 0.001 by chi-square test). In multivariable ordinal logistic regression analyses (Table 2), lacune (adjusted odds ratio 1.69, 95% confidence interval 1.16–2.47, P = 0.006), CMBs (adjusted odds ratio 1.94, 95% confidence interval 1.31–2.88, P = 0.001), and WMH (adjusted odds ratio 1.86, 95% confidence interval 1.24–2.80, P = 0.003) were independently associated with 3-month mRS scores. However, EPVS was not associated with 3-month mRS scores. When the total SVD score was entered as a dependent variable, 2, 3, and 4 points of total SVD scores were independently associated with unfavorable outcome compared with 1 point of total SVD score. Adjusted odds ratios (95% confidence interval) for 2, 3, and 4 points of total SVD scores were 1.68 (1.02–2.76), 2.24 (1.25–4.00), and 2.00 (1.02–4.29), respectively.

## Discussion

In the present study, we found an independent association between total SVD scores and functional outcome at three months following ischemic stroke. Patients with total SVD scores of 3

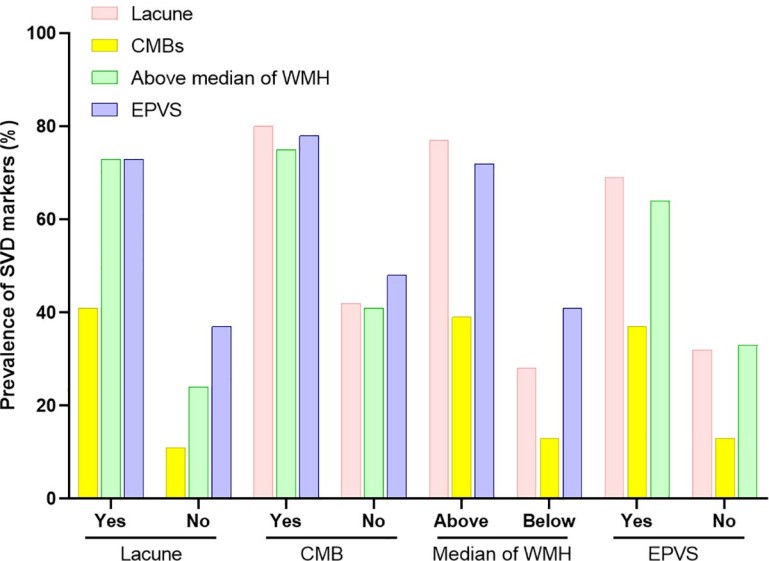

**Fig 1. Associations between markers of cerebral small vessel disease.** All p values < 0.05 (chi-square test). CMB indicates cerebral microbleed; WMH, white matter hyperintensity; EPVS, enlarged perivascular space; SVD, small vessel disease.

and 4 had a two-fold increased risk of higher mRS score at three months after ischemic stroke compared with those with a total SVD score of 0. In addition, we found among SVD markers, lacune, CMBs, and WMHs are independently associated with functional outcome. However, EPVS was not associated with functional outcome at 3-month after ischemic stroke.

In line with previous studies [5, 6, 9], we found that lacunes, CMBs, and WMHs are independently associated with unfavorable functional outcome at 3-months. These SVD markers are associated with early neurological deterioration and recurrence of stroke [6, 20]. Moreover, extensive WMH renders the brain to be more vulnerable to ischemic injury and expansion of ischemic stroke [21]. Also, patients with severe WMH or lacunes may be physically inactive

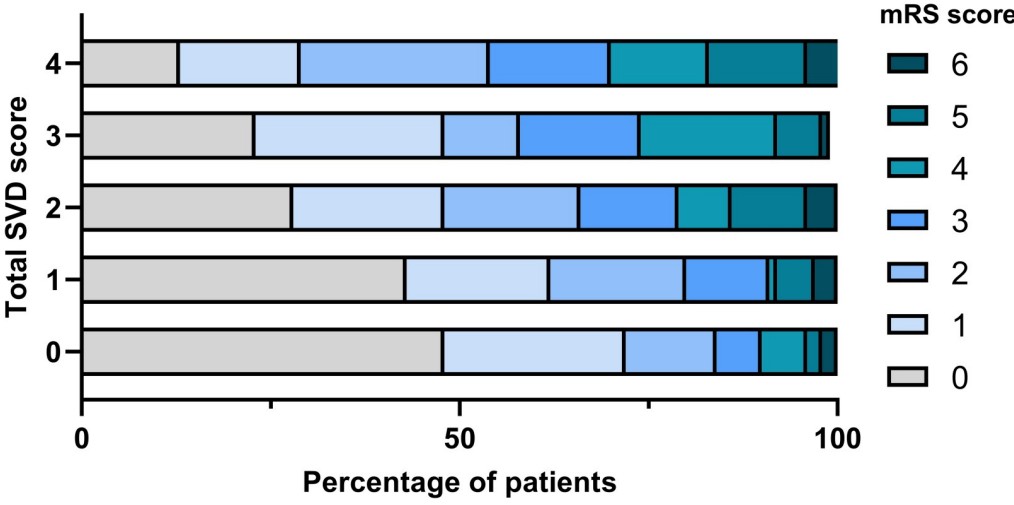

**Fig 2. Distribution of modified Rankin Scale score at three months.** p < 0.001 by chi-square test. SVD indicates small vessel disease; mRS, modified Rankin Scale.

**Table 2. Impact of total small vessel disease score on modified Rankin Scale scores at three months.**

|  | Adjusted odds ratio (95% confidence interval) | p value |
|---|---|---|
| Total SVD score |  |  |
| 0 | Reference |  |
| 1 | 0.96 (0.58–1.58) | 0.87 |
| 2 | 1.73 (1.05–2.83) | 0.031 |
| 3 | 2.18 (1.22–3.88) | 0.008 |
| 4 | 2.21 (1.04–4.69) | 0.038 |
| p for trend | 0.008 |  |
| Presence of lacunes | 1.63 (1.13–2.38) | 0.009 |
| Presence of CMBs | 1.88 (1.28–2.76) | 0.001 |
| Presence of EPVS | 1.19 (0.84–1.70) | 0.32 |
| Above median of WMH volume* | 1.87 (1.25–2.81) | 0.002 |

Adjusted for age, sex, initial National Institute Health stroke scale(NIHSS) scores, pre-stroke mRS score, previous history of stroke, hypertension, diabetes, smoking, coronary artery disease, stroke subtypes, and revascularization. Multivariable ordinal logistic regression analysis was performed for each dependent variable.

* Median = 0.85% of total brain volume

SVD indicates small vessel disease; CMB, cerebral microbleed; EPVS, enlarged perivascular space; WMH, white matter hyperintensity.

and cognitively impaired, which hampers active rehabilitation and functional regain after ischemic stroke [6].

In the present study, we found no association between EPVS and functional outcome after ischemic stroke. Although several studies have reported on the relation of EPVS with incident stroke, dementia, and recurrence after ischemic stroke [22–24], a recent study showed no association between EPVS and stroke outcome [14]. These data suggest that EPVS may have a different clinical impact on the brain with ischemic stroke from other neuroimaging markers of SVD, although EPVS has similar risk factors with other SVD markers. Indeed, we also found a positive relation between EPVS and other markers of SVD; patients with EPVS had more prevalent lacunes, CMBs, and WMH compared with those without EPVS. However, further studies are needed to delineate the clinical implications of EPVS in patients with ischemic stroke.

Neuroimaging markers of SVD are frequently observed together in patients with ischemic stroke [14]. Hence, it is helpful to quantify the total SVD burden to evaluate the cumulative effect of SVD in patients with ischemic stroke. In the present study, we found the cumulative effect of SVD on functional outcome after ischemic stroke. Compared to the effect of a single SVD marker, the effect of the quantified total SVD score was found to be greater. In the present study, the effects of total SVD scores of 3 and 4 were comparable. This may be a result from the lack of association between EPVS and stroke outcome, and thus a total SVD scores 3 or 4 could be a reasonable point to discriminate the group of a high risk of unfavorable outcome.

Several limitations of our study deserve comment. First, patients with severe stroke and revascularization therapy may have been excluded from the study. As a result, this selection bias may underestimate the degree of association between total SVD burden and function outcomes because SVD is associated with functional outcomes in patients who received thrombectomy [25] and intravenous recombinant tissue plasminogen activator [12]. Second, data from a single center and single ethinicity may have limited generalizability. Third, because the present study included only patients with MRI, our results have limited generalizability to patients who undergo only brain computed tomography (CT). Fourth, a total SVD score did

not contain information on the severity of each SVD phenotype. Irrespective of the number of lacunes, the presence of lacune awards one point. Future SVD scoring system that includes information on the severity of each marker of SVD may be useful.

In conclusion, we found an independent association between total SVD score and functional outcome at 3-month after ischemic stroke. Compared with a single marker of SVD, the total SVD score may provide more comprehensive information and improve personalized care for ischemic stroke patients.

## Acknowledgments

We thank Dr. Jung E Park for assistance with English language editing.

## Author Contributions

**Conceptualization:** Wi-Sun Ryu.

**Data curation:** Wi-Sun Ryu, Sang-Wuk Jeong, Dong-Eog Kim.

**Formal analysis:** Wi-Sun Ryu, Dong-Eog Kim.

**Funding acquisition:** Wi-Sun Ryu.

**Investigation:** Wi-Sun Ryu.

**Methodology:** Wi-Sun Ryu, Sang-Wuk Jeong.

**Resources:** Sang-Wuk Jeong, Dong-Eog Kim.

**Supervision:** Wi-Sun Ryu, Dong-Eog Kim.

**Writing – original draft:** Wi-Sun Ryu.

**Writing – review & editing:** Sang-Wuk Jeong, Dong-Eog Kim.

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
