## [Decision Letter · Decision Letter 0]

9 Oct 2020

PONE-D-20-28449

Total Small Vessel Disease Burden and Functional Outcome in Patients with Ischemic Stroke

PLOS ONE

Dear Dr. Ryu,

Thank you for submitting your manuscript to PLOS ONE. After careful consideration, we feel that it has merit but does not fully meet PLOS ONE’s publication criteria as it currently stands. Therefore, we invite you to submit a revised version of the manuscript that addresses the points raised during the review process.

Question about the reason for not adjusting the multivariate model with the NIHSS score.English style needs some editing.Provide more detail information based on reviewer’s suggestions.

We look forward to receiving your revised manuscript.

Kind regards,

Quan Jiang, Ph,D.

Academic Editor

PLOS ONE

Journal Requirements:

2.We note that you have indicated that data from this study are available upon request. PLOS only allows data to be available upon request if there are legal or ethical restrictions on sharing data publicly. For information on unacceptable data access restrictions, please see http://journals.plos.org/plosone/s/data-availability#loc-unacceptable-data-access-restrictions.

3.Thank you for stating the following financial disclosure:

 [WSR: This work was supported by the Dongguk University Research Program. The funders had no role in study design, data collection and analysis, decision to publish, or preparation of the manuscript.].

We note that one or more of the authors is affiliated with the funding organization, indicating the funder may have had some role in the design, data collection, analysis or preparation of your manuscript for publication; in other words, the funder played an indirect role through the participation of the co-authors. If the funding organization did not play a role in the study design, data collection and analysis, decision to publish, or preparation of the manuscript and only provided financial support in the form of authors' salaries and/or research materials, please do the following:

Review your statements relating to the author contributions, and ensure you have specifically and accurately indicated the role(s) that these authors had in your study. These amendments should be made in the online form.

Confirm in your cover letter that you agree with the following statement, and we will change the online submission form on your behalf:

Reviewers' comments:

Reviewer's Responses to Questions

**Comments to the Author**

1. Is the manuscript technically sound, and do the data support the conclusions?

Reviewer #1: Yes

Reviewer #2: Partly

2. Has the statistical analysis been performed appropriately and rigorously? 

Reviewer #1: No

Reviewer #2: No

3. Have the authors made all data underlying the findings in their manuscript fully available?

Reviewer #1: Yes

Reviewer #2: Yes

4. Is the manuscript presented in an intelligible fashion and written in standard English?

Reviewer #1: Yes

Reviewer #2: Yes

5. Review Comments to the Author

Reviewer #1: Very interesting paper where the authors use a validated score to evaluate the role of cerebral small vessel disease on stroke outcome.

Major concerns

Could please the authors explain the reason for not adjusting the multivariate model with theNIHSS score?. Has this analysis been performed?. It is a common predictor of functional outcome in patients with stroke and patients with SVD 4 tend to have higher NIHSS scores (p75 is 10).

Minor concerns

Some of the English writing should be reviewed.

The authors use many times in the manuscript the expression “different subtypes of SVD” when they actually refer to different neuroimaging markers of SVD. I suggest use the expressions “SVD markers” or “neurimaging markers of SVD” instead. The “use of subtypes of SVD” can cause confusion to the readers, since SVD has several underlying etiologies such as hypertensive, amyloid angiopathy, CADASIL.. and the expression “SVD subtypes” is frequently used in this setting.

The authors perform a multivariate ordinal regression analysis. Therefore, it would be more correct if they say that “lacunes, MBS… independently increased the risk of worsening on the mRS scale” instead of “lacunes, MBS….were independently associated with unfavorable outcome”, because they analyze the whole spectrum of the scale and not only the unfavorable outcome categories(3-6).

Why have the authors selected the period 2012 to 2013 for the analysis of 90 days mRS?. Don’t they have more recent data?

How many centers participate in the study?. It is not clear from the methods section. At the beginning of this section, the authors say”patients who visited our center”, but lately they say “stroke subtypes were determined by the consensus experienced neurologists in each participating center”.

Reviewer #2: In this study authors evaluated the association between cerebral small vessel disease and functional outcome after ischemic stroke. The study is confirmative, since this observation is not novel in patients with ischemic stroke, but may be of interest because all patients had MR examination and SVD was therefore quantified with the gold standard. Furthermore, the study has been conducted within an Asian cohort of patients whereas many study on the topic were with Caucasian patients, therefore confirms the negative role of SVD on stroke outcome independently of the ethnicity.

The study could be improved and English style needs some editing.

Comments:

- Please report in the abstract number and strength of association rather than p values

- Please present general characteristics of population in the first part of results (age, sex, median NIHSS, etc.)

- Please state inclusion and exclusion criteria for study population. E.g. were included patients treated with acute stroke treatment?

- Please move the excluded patients paragraph from methods to results section

- a single reader rated all the scans

- Table 1 needs some editing (e.g. age, years, probably “mean” is missing). Please add a column with data from all the population.

- What “revascularization” means in the adjusted analysis?

- Although stroke severity (i.e. NIHSS) is a major determinant of stroke outcome, NIHSS is missing in the adjusted analysis. In table 1 there is a clear trend towards worse NIHSS in patients with higher SVD burden. Conversely, in the adjusted analysis

there are some variables that likely do not affect stroke outcome (e.g. smoke exposure, coronary heart disease-anamnestic, I suppose-). I would like to see a further multivariable analysis adjusted simply for age, sex, NIHSS, pre-stroke mRS.

- Figure 1 is rather confusing

- Discussion: there are also reports about no association between SVD and functional outcome after stroke (e.g. Boulouis G et al., Neurology 2019), please discuss your results comparing also with such studies.

- Discussion: as a limit, authors mentioned that patients that were eligible for endovascular treatment were excluded from the study. However, there are reports about SVD and outcomes in such patients (Arba F et al. Neurol Sci 2019; Boulouis G et al., Neurology 2019). Again, what about patients eligible for intravenous rt-PA (see Charidimou A et al., Stroke 2016; IST-3 collaborators, Lancet Neurol; Arba F et al., Acta Neurol Scand 2017)?

- Results apply only to patients who underwent MR, but the great majority of stroke patients receive only CT. Please discuss.

6. PLOS authors have the option to publish the peer review history of their article (what does this mean?). If published, this will include your full peer review and any attached files.

Reviewer #1: **Yes: **Manuel Gomez-Choco

Reviewer #2: **Yes: **Francesco Arba

---

## [Author Response · Author response to Decision Letter 0]

20 Oct 2020

We thank the reviewers for taking time. Please refer the attached file "Response to reviewers' comment."

---

## [Decision Letter · Decision Letter 1]

2 Nov 2020

Total Small Vessel Disease Burden and Functional Outcome in Patients with Ischemic Stroke

PONE-D-20-28449R1

Dear Dr. Ryu,

We’re pleased to inform you that your manuscript has been judged scientifically suitable for publication and will be formally accepted for publication once it meets all outstanding technical requirements.

Kind regards,

Quan Jiang, Ph,D.

Academic Editor

PLOS ONE

Additional Editor Comments (optional):

Reviewers' comments:

Reviewer's Responses to Questions

**Comments to the Author**

1. If the authors have adequately addressed your comments raised in a previous round of review and you feel that this manuscript is now acceptable for publication, you may indicate that here to bypass the “Comments to the Author” section, enter your conflict of interest statement in the “Confidential to Editor” section, and submit your "Accept" recommendation.

Reviewer #1: All comments have been addressed

Reviewer #2: All comments have been addressed

2. Is the manuscript technically sound, and do the data support the conclusions?

Reviewer #1: (No Response)

Reviewer #2: Yes

3. Has the statistical analysis been performed appropriately and rigorously? 

Reviewer #1: (No Response)

Reviewer #2: Yes

4. Have the authors made all data underlying the findings in their manuscript fully available?

Reviewer #1: (No Response)

Reviewer #2: Yes

5. Is the manuscript presented in an intelligible fashion and written in standard English?

Reviewer #1: (No Response)

Reviewer #2: Yes

6. Review Comments to the Author

Reviewer #1: (No Response)

Reviewer #2: No further comments, all issues have been addressed. The manuscript sounds more complete and limitations are fairly discussed.

7. PLOS authors have the option to publish the peer review history of their article (what does this mean?). If published, this will include your full peer review and any attached files.

Reviewer #1: **Yes: **Manuel Gómez-Choco

Reviewer #2: **Yes: **Francesco Arba

---

## [Editor Report · Acceptance letter]

4 Nov 2020

PONE-D-20-28449R1 

Total Small Vessel Disease Burden and Functional Outcome in Patients with Ischemic Stroke 

Dear Dr. Ryu:

I'm pleased to inform you that your manuscript has been deemed suitable for publication in PLOS ONE. Congratulations! Your manuscript is now with our production department. 

Kind regards, 

on behalf of

Dr. Quan Jiang 

Academic Editor

PLOS ONE